# Dual Distillation of Trajectory and Guidance Knowledge for Faster Inference in Conditional Masked Diffusion Language Models

## Abstract

Masked diffusion language models (MDLMs) have emerged as a promising generative framework for natural language, owing to parallel non-autoregressive generation capabilities with iterative unmasking/denoising. However, typical MDLMs require a very large number of neural network function evaluations for effective inference, making them computationally expensive in many real-world NLP applications that rely on *conditional* sequence-to-sequence generation. In this work, we propose a *two-stage distillation* method for conditional MDLMs that distills knowledge of (i) *classifier-free guidance* as well as (ii) *unmasking trajectory* from the existing teacher MDLM into a student MDLM. This allows the student MDLM, during inference, to (i) reduce two forward passes, required by a classifier-free guided (teacher) MDLM, to a single pass, and (ii) drastically reduce the number of unmasking steps. In this way, by dual distillation of guidance and trajectory knowledge, our MDLM achieves *speedups of up to 16×* while virtually retaining the quality of generation.

## 1 Introduction

The rapid progress in deep learning–based language models, particularly autoregressive large language models (LLMs) (Dubey et al., 2024; Yang et al., 2025) and, more recently, masked diffusion language models (MDLMs) (Nie et al., 2025b; Zhao et al., 2025), has led to their widespread deployment across diverse applications, including conversational agents, code generation, and machine translation. Given their adoption at both small scale (task-specific models) and large scale (general-purpose systems), efficiency during deployment has emerged as a critical requirement. To this end, a variety of techniques have been proposed to optimize model usage while preserving performance. At the training stage, knowledge distillation has been extensively studied as a means to transfer knowledge from large pre-trained models to more compact students (Gu et al., 2024; Agarwal et al., 2024). At the inference stage, approaches such as KV-caching (Li et al., 2025) and speculative decoding (Leviathan et al., 2023) have demonstrated significant improvements in runtime efficiency for LLMs. Parallel efforts are now underway for MDLMs, with recent studies exploring KV-caching (Ma et al., 2025; Hu et al., 2025) and distillation strategies (Deschenaux & Gulcehre, 2025; Hayakawa et al., 2025) to enable efficient large-scale deployment.

Prior work on improving the generation efficiency of diffusion models for image synthesis through distillation has identified two primary sources of inefficiency: **(1)** the requirement of multiple iterative steps to generate outputs (Salimans & Ho, 2022), and **(2)** the need for two forward passes through the denoising network when applying classifier-free guidance (Meng et al., 2023). Distillation in the first case aims to reduce the number of sampling steps necessary to achieve high-quality outputs, while in the second, it seeks to eliminate the overhead introduced by dual forward passes. For masked diffusion language models (MDLMs), prior studies have similarly investigated distillation to reduce the number of generation steps; however, these efforts have been restricted to unconditional generation and sentence completion tasks. This narrow scope limits our understanding of distillation for MDLMs, as the sampling inefficiencies of guided MDLMs remain largely unexplored, particularly in the context of sequence-to-sequence task-specific settings.

Our aim in this work is to develop efficient conditional MDLMs for sequence-to-sequence NLP tasks via the framework of knowledge distillation. Our contributions are

1. a two-stage distillation framework for conditional MDLMs, which trains student models to tackle sampling inefficiencies that arise due to classifier-free guidance and multi-step denoising; to the best of our knowledge, our work is the first to explore distillation for guided conditional MDLMs for seq-to-seq NLP tasks.

2. comprehensive analysis on multiple seq-to-seq NLP tasks showcasing improved efficiency versus performance trade-off in the distilled MDLMs compared to the vanilla-fine-tuned MDLMs; our presented distilled MDLMs exhibit up to $16\times$ speedup without significant degradation in sample quality, as well as improve few-step generation.

## 2 PRELIMINARIES

### 2.1 MASKED DIFFUSION LANGUAGE MODELS

Masked diffusion language models (MDLMs) define the diffusion forward and reverse processes in the discrete (or the token) space (Hoogeboom et al., 2021; Austin et al., 2021). The forward process iteratively replaces tokens in the sequence with mask tokens, and the reverse process is learned to iteratively unmask tokens to generate new samples (Sahoo et al., 2024; Nie et al., 2025a).

Let $V$ be a fixed word/token vocabulary. Let $t \in [0, 1]$ be the timestep for the diffusion process, where $t = 0$ corresponds to no noise and $t = 1$ corresponds to a fully masked sequence. We denote a ground truth sequence of length $L$ from a given dataset as $\boldsymbol{x}_0 = (x_0^{(0)}, x_0^{(1)}, x_0^{(2)}, ...., x_0^{(L-1)})$ where $\forall i \in \{0, 1, ..., L-1\}, x_0^{(i)} \in V$. The forward process is defined as

$$q_{t|0}(\mathbf{x}_t|\boldsymbol{x}_0) = \prod_{i=0}^{L-1} q_{t|0}(\mathbf{x}_t^{(i)}|x_0^{(i)}) \quad \text{where} \quad q_{t|0}(\mathbf{x}_t^{(i)}|x_0^{(i)}) := \begin{cases} \alpha_t, & \mathbf{x}_t^{(i)} = x_0^{(i)}, \\ 1 - \alpha_t, & \mathbf{x}_t^{(i)} = \langle m \rangle, \end{cases} \tag{1}$$

where $\langle m \rangle$ represents the mask token, $\mathbf{x}_t$ is the vector of discrete random variables over the set $V \cup \{\langle m \rangle\}$, representing the partially-masked sequence at timestep $t$, $\alpha_t$ is the hyperparameter controlling the proportion of masked tokens in the sequence at a given timestep, and $q_0(\cdot)$ is the data distribution. To reverse this process for generation, we train a denoising neural network with parameters $\theta$ that outputs $p_\theta(\cdot|\boldsymbol{x}_t)$ estimating the true data distribution given the noisy sequence (i.e., $q_{0|t}(\cdot|\boldsymbol{x}_t)$)[1]. For timesteps $s$ and $t$ with $0 \leq s < t \leq 1$, the reverse process is obtained as:

$$p_\theta(\mathbf{x}_s|\boldsymbol{x}_t) = \prod_{i=0}^{L-1} p_\theta(\mathbf{x}_s^{(i)}|\boldsymbol{x}_t) \quad \text{where}$$

$$p_\theta(\mathbf{x}_s^{(i)}|\boldsymbol{x}_t) = \begin{cases} \frac{t-s}{t} p_\theta(\mathbf{x}_0^{(i)}|\boldsymbol{x}_t), & x_t^{(i)} = \langle m \rangle, \mathbf{x}_s^{(i)} \neq \langle m \rangle \\ \frac{s}{t}, & x_t^{(i)} = \langle m \rangle, \mathbf{x}_s^{(i)} = \langle m \rangle \\ 1, & x_t^{(i)} \neq \langle m \rangle, \mathbf{x}_s^{(i)} = x_t^{(i)}. \end{cases} \tag{2}$$

Training uses a simplified upper bound on the negative log-likelihood as the objective (Shi et al., 2024; Sahoo et al., 2024).

$$-\log p_\theta(\boldsymbol{x}_0) \leq \int_0^1 \frac{\alpha_t'}{1 - \alpha_t} \mathop{\mathbb{E}}_{q(\mathbf{x}_t|\boldsymbol{x}_0)} \left[ \sum_{\forall i : x_t^i = \langle m \rangle} -\log p_\theta(\mathbf{x}_0^i|\boldsymbol{x}_t) \right] dt. \tag{3}$$

Here, $\alpha_t'$ is the time derivative of $\alpha_t$. Unlike typical diffusion models, masked diffusion processes can be learned with a time-independent neural network, which allows the use of standard Transformer-based architectures without extra time conditioning to parameterize the reverse process (Zheng et al., 2025).

---

[1]This is equivalent to estimating the token distribution on the masked positions given the unmasked tokens.

## 2.2 KNOWLEDGE DISTILLATION IN LANGUAGE MODELS

Knowledge distillation (Hinton et al., 2015) refers to a broad spectrum of techniques that make use of a capable teacher model to train a more efficient student model. The idea is to train a student model to mimic outputs from a more complex teacher model. We particularly focus on distillation techniques that enable student models to mimic output distributions from a teacher model using divergence-based loss functions. In language modeling, earlier works have successfully explored such techniques on masked language models (Sanh et al., 2019) as well as generative encoder-decoder and decoder-only language models (Wen et al., 2023). Recent works have also applied reinforcement learning-based divergence minimization for distilling autoregressive LLMs (Gu et al., 2024; Agarwal et al., 2024).

## 3 DISTILLATION OF CONDITIONAL MASKED DIFFUSION LANGUAGE MODELS

In this section, we introduce distillation techniques for training efficient conditional MDLMs on seq-to-seq tasks. Let the conditional sequence be denoted as $\boldsymbol{y} = (y^{(0)}, y^{(1)}, \ldots, y^{(L_1-1)})$ and the ground truth target sequence be denoted as $\boldsymbol{x}_0 = (x_0^{(0)}, x_0^{(1)}, \ldots, x_0^{(L_2-1)})$. The diffusion process operates on the target sequence, where we denote $\boldsymbol{x}_t$ as a noisy (or partially masked) sequence at time $t$. Let $\widehat{\mathbf{x}}_\theta(\boldsymbol{x}_t, \boldsymbol{y})$ be a masked diffusion denoiser with parameters $\theta$ that estimates the log-probabilities of the clean sequence given $\boldsymbol{x}_t$ and $\boldsymbol{y}$ (i.e., $\widehat{\mathbf{x}}_\theta(\boldsymbol{x}_t, \boldsymbol{y}) \approx \log p_\theta(\mathbf{x}_0|\boldsymbol{x}_t, \boldsymbol{y})$). In practice, the input to the network is passed by concatenating $\boldsymbol{y}$ and $\boldsymbol{x}_t$ with a separator token or a string in between.

We improve the inference-time efficiency by aiming to reduce the **number of function evaluations** (**NFEs**, which is equal to the number of forward passes from the denoiser) required to generate target sequences without significant degradation in generation quality. We first fine-tune an MDLM for a specific seq-to-seq task using a standard training method following Nie et al. (2025a) (refer to Algorithm 4 in Appendix B for the details). Next, we propose a two-stage distillation procedure (similar to Meng et al. (2023)). In the first stage, we distill guided teacher outputs into a student model. In the second stage, we distill multi-step outputs from the distilled model obtained from the first stage into another student model. In the following subsections, we discuss both stages in detail.

### 3.1 FIRST STAGE: GUIDANCE DISTILLATION

Classifier-free guidance (CFG) allows for improving grounding on the given condition by trading off generation diversity (Ho & Salimans, 2021). The classifier-free guided log-probabilities of the true data $\mathbf{x}_0$ are computed as follows (Schiff et al., 2025):

$$\log p^\gamma(\mathbf{x}_0|\boldsymbol{x}_t, \boldsymbol{y}) \propto \gamma \log p(\mathbf{x}_0|\boldsymbol{x}_t, \boldsymbol{y}) + (1 - \gamma) \log p(\mathbf{x}_0|\boldsymbol{x}_t) \tag{4}$$

Here, $\gamma$ is the guidance scale which controls the diversity versus fidelity trade-off. To compute the unconditional log-probabilities (i.e., the second term in the RHS of Equation 4) using the denoiser, we replace $\boldsymbol{y}$ with $\phi = \langle m \rangle_{|\boldsymbol{y}|}$, which is a sequence of all masked tokens of the same length as $\boldsymbol{y}$. Although prior works have proven that using the above formulation consistently improves generation quality as long as the guidance scale is properly tuned, the computation requires the evaluation of both conditional (i.e., $\log p(\mathbf{x}_0|\boldsymbol{x}_t, \boldsymbol{y})$) and unconditional log-probabilities (i.e., $\log p(\mathbf{x}_0|\boldsymbol{x}_t)$). This doubles the NFEs required at every denoising step. In this stage, we tackle this inefficiency by distilling classifier-free guided outputs from the teacher model into a single forward pass of a student model.

We distill guided outputs on a range of guidance scales to retain the flexibility of choosing the right trade-off between diversity and fidelity. To do so, we need to incorporate an additional scalar input (i.e., guidance scale) into the student denoiser. As mentioned in Section 2.1, we use a standard Transformer architecture as our denoiser. To this, we first add an extra two-layer MLP with SiLU activation function (Elfwing et al., 2017). The guidance scale value is converted to a sinusoidal embedding and passed through the MLP, producing an embedding of the same size as the hidden

---

**Algorithm 1:** Distilling Guidance Knowledge for Conditional MDLMs

**Given:** Trained teacher denoiser $\widehat{\mathbf{x}}_\eta(\cdot, \boldsymbol{y})$, untrained student denoiser $\widehat{\mathbf{x}}_\theta(\cdot, \boldsymbol{y}, \gamma)$, guidance scale range $[\gamma_{\min}, \gamma_{\max}]$, mask token $\langle m \rangle$.

$\theta \leftarrow \eta$ ; ▷ Initialize student with teacher weights

**repeat**

  $(\boldsymbol{y}, \boldsymbol{x}_0) \sim \text{data}, t \sim \mathcal{U}(0, 1), \boldsymbol{x}_t \sim q_{t|0}(\mathbf{x}_t | \boldsymbol{x}_0)$ ; ▷ forward diffusion process

  $\phi \leftarrow \langle m \rangle_{|\boldsymbol{y}|}$ ; ▷ sequence of $\langle m \rangle$ tokens of length $|\boldsymbol{y}|$

  $\gamma \sim \mathcal{U}(\gamma_{\min}, \gamma_{\max})$; ▷ sample guidance scale

  $\log p_{\text{teacher}} \leftarrow \gamma \widehat{\mathbf{x}}_\eta(\boldsymbol{x}_t, \boldsymbol{y}) + (1 - \gamma) \widehat{\mathbf{x}}_\eta(\boldsymbol{x}_t, \phi)$ ; ▷ CFG from teacher

  $\log p_{\text{student}} \leftarrow \widehat{\mathbf{x}}_\theta(\boldsymbol{x}_t, \boldsymbol{y}, \gamma)$ ; ▷ single forward pass of the student

  $\mathcal{L}_{\text{GD}} = \sum\limits_{\forall i: x_t^{(i)} = \langle m \rangle} D_{\text{KL}}(p_{\text{student}}^{(i)} \| p_{\text{teacher}}^{(i)})$ ; ▷ Compute loss

  Gradient descent with $\nabla_\theta \mathcal{L}_{\text{GD}}$;

**until** *convergence*;

---

size of the Transformer. The embedding is then prepended to the token embeddings of the input. The self-attention mechanism conditions the sequence on the CFG-scale embedding, which gets jointly learnt with the distillation. We denote this modified student network with parameters $\theta$ as $\widehat{\mathbf{x}}_\theta(\cdot, y, \gamma)$.

Algorithm 1 shows the proposed guidance distillation algorithm for conditional MDLMs. Let $\widehat{\mathbf{x}}_\theta$ and $\widehat{\mathbf{x}}_\eta$ be the student and teacher MDLMs, respectively. The parameters of the student denoiser $\widehat{\mathbf{x}}_\theta$ are initialized with the teacher weights $\eta$ except for the MLP parameters, where the first layer is initialized randomly and the second layer is initialized with zeros. We first sample a guidance scale value $\gamma$ uniformly from a given range $[\gamma_{\min}, \gamma_{\max}]$. Next, we compute a CFG estimate of the log-probabilities of the target sequence from the teacher model following Equation 4 (i.e., $\gamma \widehat{\mathbf{x}}_\eta(\boldsymbol{x}_t, \boldsymbol{y}) + (1 - \gamma) \widehat{\mathbf{x}}_\eta(\boldsymbol{x}_t, \phi)$) and a conditional estimate of the target sequence from the student model (i.e., $\widehat{\mathbf{x}}_\theta(\boldsymbol{x}_t, \boldsymbol{y}, \gamma)$). To push the conditional estimate of the student closer to the guided estimate of the teacher, we employ the mode-seeking KL-divergence loss $D_{\text{KL}}(p_\theta^\gamma \| p_\eta^\gamma)$. The loss is computed over all masked positions in $\boldsymbol{x}_t$.

### 3.2 SECOND STAGE: PROGRESSIVE TRAJECTORY DISTILLATION

In this stage, we employ distillation to obtain a student model with improved few-step generation quality. Our base method follows the progressive distillation setup called self-distillation through time (SDTT) as proposed in Deschenaux & Gulcehre (2025). We adapt the setup for conditional MDLMs and incorporate distillation from a guidance distilled MDLM (obtained from the first stage as described in Section 3.1).

Algorithm 2 describes the trajectory distillation mechanism that enables better few-step sampling. We begin with the guidance distilled MDLM $\widehat{\mathbf{x}}_\theta(\cdot, \boldsymbol{y}, \gamma)$. Let the new student model to be trained be denoted as $\widehat{\mathbf{x}}_\xi(\cdot, \mathbf{y}, \gamma)$. The core idea is to mimic outputs from the teacher that are generated with multiple sampling steps into a single step from a student model. We consider a special case of this scenario where we distill **two steps** from the teacher with the fixed step size into a single student step (Salimans & Ho, 2022). To begin, we first fix an initial sampling step size $\Delta(\leq 1)$. Given the guidance scale $\gamma \sim \mathcal{U}(\gamma_{\min}, \gamma_{\max})$, a function $\texttt{solve}(\boldsymbol{x}_t, \Delta, \widehat{\mathbf{x}}_\theta(\cdot, \boldsymbol{y}, \gamma))$ is defined that takes $\boldsymbol{x}_t$ and first samples $\boldsymbol{x}_{t-2\Delta}$ by simulating **two reverse diffusion steps** each of step size $\Delta$ using Equation 2. We gather the log-probabilities on the unmasked positions in each step during this process. Finally, the function takes an additional step to compute log-probabilities on all the remaining masked positions in $\boldsymbol{x}_{t-2\Delta}$ and returns the target log-probabilities. More details on $\texttt{solve}$ appear in Appendix A and Algorithm 3. To train the student denoiser $\mathbf{x}_\xi$, similar to guidance distillation from Section 3.1, we compute the mode-seeking KL-divergence loss on all masked positions of $\boldsymbol{x}_t$.

We progressively repeat this distillation after fixed training intervals by replacing the teacher with the student and doubling the step size. This allows the student model to be progressively trained to better mimic the teacher's estimate of the data distribution with larger and larger step sizes (essentially reducing the number of sampling steps).

---

**Algorithm 2:** Distilling Trajectory Knowledge for Conditional MDLMs

---

**Given:** Trained teacher denoiser $\widehat{\mathbf{x}}_\theta(\cdot, \boldsymbol{y}, \gamma)$, untrained student denoiser $\widehat{\mathbf{x}}_\xi(\cdot, \boldsymbol{y}, \gamma)$, guidance scale range $[\gamma_{\min}, \gamma_{\max}]$, number of distillation rounds $R$, sampling step size $\Delta$.

**Given:** $\mathtt{solve}(\boldsymbol{x}_t, \Delta, \widehat{\mathbf{x}}_\theta(\cdot, \boldsymbol{y}, \gamma))$ that uses $\widehat{\mathbf{x}}_\theta$ to unmask $\boldsymbol{x}_t$ for two sampling steps with $\Delta$ step size and one extra step to unmask the remaining masked positions (refer to Algorithm 3).

$\xi \leftarrow \theta$ ;           ▷ Initialize student with teacher's weights

**for** $R$ *iterations* **do**
    **repeat**
        $(\boldsymbol{y}, \boldsymbol{x}_0) \sim \text{data}, t \sim \mathcal{U}(0,1), \boldsymbol{x}_t \sim q_{t|0}(\mathbf{x}_t|\boldsymbol{x}_0)$ ;    ▷ forward diffusion process
        $\gamma \sim \mathcal{U}(\gamma_{\min}, \gamma_{\max})$;           ▷ sample guidance scale
        $\log p_{\text{teacher}} \leftarrow \mathtt{solve}(\boldsymbol{x}_t, \Delta, \widehat{\mathbf{x}}_\theta(\cdot, \boldsymbol{y}, \gamma))$;     ▷ simulate two sampling steps
        $\log p_{\text{student}} \leftarrow \widehat{\mathbf{x}}_\xi(\boldsymbol{x}_t, \boldsymbol{y}, \gamma)$ ;      ▷ single forward pass of the student
        $\mathcal{L}_{\text{TD}} = \sum\limits_{\forall i:\, x_t^{(i)} = \langle m \rangle} D_{\text{KL}}(p_{\text{student}}^{(i)} \| p_{\text{teacher}}^{(i)})$ ;          ▷ Compute loss
        Gradient descent with $\nabla_\xi \mathcal{L}_{\text{TD}}$
    **until** *convergence*;
    $\theta \leftarrow \xi$;           ▷ Student becomes the new teacher
    $\Delta \leftarrow \Delta * 2$;           ▷ Double the step size
**end**

---

# 4 EXPERIMENTAL SETTINGS

## 4.1 DATASETS AND EVALUATION

We evaluate our distillation approach on three sequence-to-sequence NLP tasks: **(1) Bible style transfer**, using the dataset of Carlson et al. (2017), which contains parallel Biblical sentences paired with simplified English and other versions of the Bible; in particular, we consider the PUB-BBE (Public Bible versions → Bible in Basic English) and PUB-ASV (Public Bible versions → American Standard Version) tasks; **(2) paraphrasing**, using the Quora Question Pairs (QQP) dataset (Sharma et al., 2019); and **(3) question generation**, using the Quasar-T dataset (Dhingra et al., 2017), following the setup of Gong et al. (2023).

For evaluation, we report METEOR (Banerjee & Lavie, 2005) and ROUGE-L (Lin, 2004), two standard $n$-gram-based metrics that compare generated outputs against reference texts, across all tasks. Efficiency is measured in terms of speedup, defined by the reduction in the number of function evaluations (NFEs), where NFE corresponds to the total number of forward passes through the MDLM during sampling. We distinguish between *denoising steps* and NFEs: for the vanilla-fine-tuned teacher MDLM, each denoising step requires two forward passes due to classifier-free guidance, yielding NFE $= 2 \times$ denoising-steps. In contrast, for the distilled MDLMs, guidance distillation reduces this computation to a single forward pass per step, resulting in NFE $=$ denoising-steps.

## 4.2 TRAINING AND INFERENCE DETAILS

To obtain initial teacher models fine-tuned on seq-to-seq tasks, we fine-tune the pre-trained MDLM with 113M non-embedding parameters released by Nie et al. (2025a)[2]. We fix the noise scheduler $\alpha_t = 1 - t$. All our training experiments are conducted on a single NVIDIA RTX A6000 GPU with *bfloat16* precision. For optimization, we use the AdamW optimizer (Loshchilov & Hutter, 2019) with $\beta_1 = 0.9$ and $\beta_2 = 0.95$. We set a small value $\epsilon = 10^{-5}$, indicating the minimum possible noise level to prevent numerical overflow in loss calculation. The learning rate is linearly warmed up to a maximum value after which it stays constant. The gradient norm is clipped to 1.0 during training. We store an exponential moving average of weights during training with a decay rate of 0.999, which is used in the inference phase. We observed that the loss curves during the distillation phase were quite unintuitive and didn't properly indicate convergence. For stage-1 (guidance) distillation, we set $\gamma_{\min} = 1$ and $\gamma_{\max} = 3$. For stage-2 (trajectory) distillation, we fix the initial step size $\Delta$ as

---

[2]https://huggingface.co/nieshen/SMDM

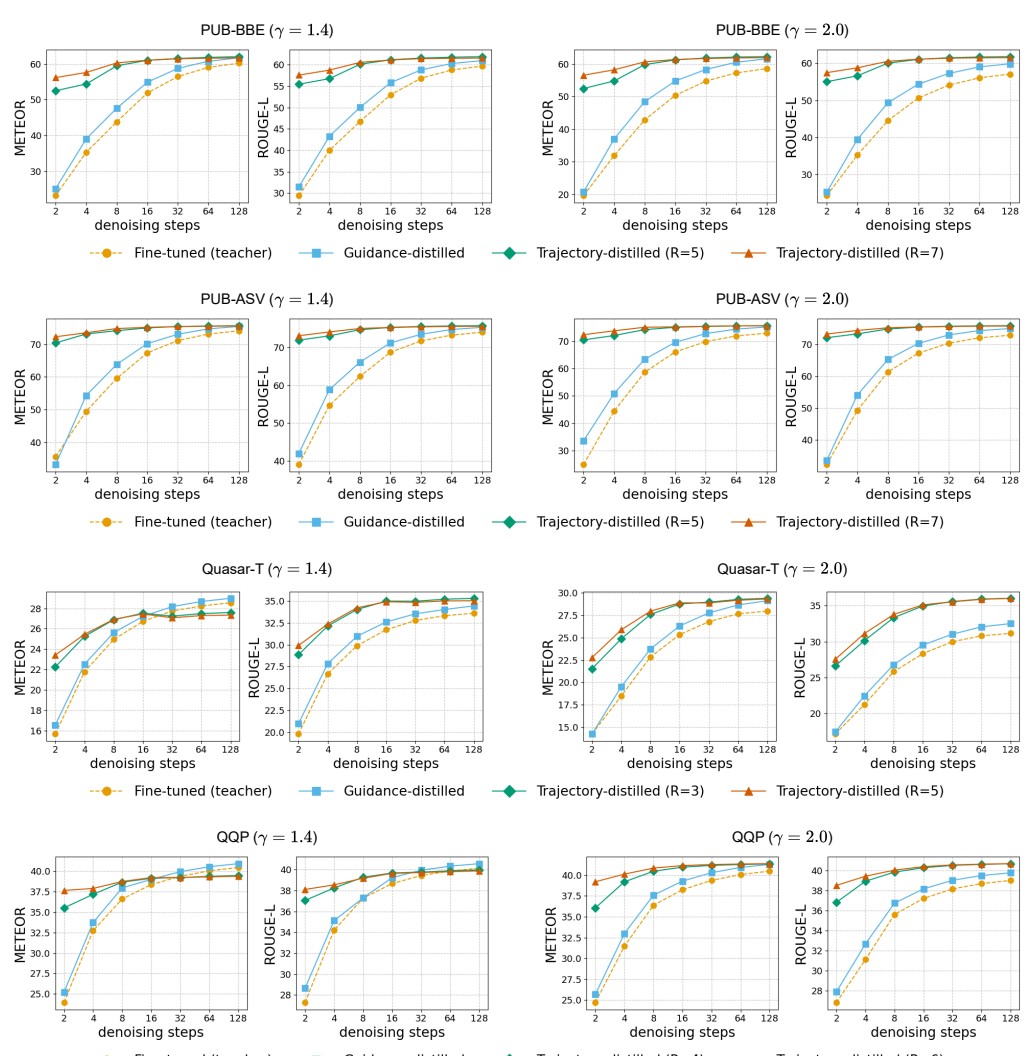

**Figure 1:** Main results displaying the metrics versus denoising steps plots for the four tasks mentioned in Section 4.1. Here, $\gamma$ refers to the guidance scale during inference and $R$ refers to the number of distillation rounds in trajectory distillation (which may differ for each task as mentioned in Section 4.2). Our distilled MDLMs consistently outperform the vanilla-fine-tuned teacher on *few-step generation*.

$1/2^k$ ($k \in \mathbb{N}$) depending on the task, which also gives us the total number of rounds $R = k$. To better assess the training progress, we generated outputs after fixed training step intervals on a small validation set and tracked the metrics mentioned in Section 4.1. For inference, we use a greedy heuristic-based sampling strategy proposed in Chang et al. (2022), details of which are present in Algorithm 5 in the Appendix B. We set the sampling temperature to 0.5. Further hyperparameter details can be found in Appendix D.

## 5 RESULTS AND DISCUSSION

Figure 1 shows the main results of our distillation experiments with the datasets mentioned in the previous section. We plot metrics versus denoising steps on every test set with three settings: **(1)** vanilla-fine-tuned MDLM on the given task, **(2)** guidance-distilled MDLM with vanilla-fine-tuned model as the teacher, and **(3)** trajectory-distilled MDLM with $R$ distillation rounds where the teacher in the first round is the guidance-distilled MDLM and the teacher in a subsequent round $r$ is the

**Table 1:** Ablations on different loss functions for guidance distillation on various seq-to-seq NLP tasks. The guidance distilled MDLMs using divergence-based loss functions (based on KL divergence and TVD) outperform the teacher MDLM (with mode-seeking KL-divergence performing the best), while the MDLM distilled with MSE loss underperforms significantly. The highlighted mode-seeking KL loss is used in the main experiments.

| Settings | Denoising steps | PUB-BBE METEOR $\gamma=1.4$ | $\gamma=2.0$ | ROUGE-L $\gamma=1.4$ | $\gamma=2.0$ | PUB-ASV METEOR $\gamma=1.4$ | $\gamma=2.0$ | ROUGE-L $\gamma=1.4$ | $\gamma=2.0$ |
|---|---|---|---|---|---|---|---|---|---|
| Vanilla-fine-tuned (teacher) | 4 | 35.368 | 31.858 | 39.997 | 35.232 | 49.449 | 44.43 | 54.657 | 49.254 |
| | 16 | 51.897 | 50.385 | 52.972 | 50.657 | 67.354 | 66.123 | 68.789 | 67.235 |
| | 64 | 59.03 | 57.293 | 58.77 | 56.102 | 73.102 | 71.987 | 73.329 | 72.027 |
| mode-seeking KL | 4 | 39.11 | 36.978 | 43.281 | 39.49 | 54.304 | 50.915 | 58.846 | 54.085 |
| | 16 | 54.893 | 54.788 | 55.791 | 54.401 | 70.176 | 69.652 | 71.282 | 70.28 |
| | 64 | 60.734 | 60.551 | 60.272 | 59.043 | 74.693 | 74.495 | 74.83 | 74.295 |
| mode-covering KL | 4 | 37.81 | 34.714 | 42.197 | 37.069 | 52.621 | 48.452 | 57.448 | 51.834 |
| | 16 | 54.245 | 53.577 | 55.417 | 53.339 | 69.24 | 68.319 | 70.521 | 69.041 |
| | 64 | 60.335 | 59.859 | 60.078 | 58.332 | 74.208 | 73.855 | 74.356 | 73.663 |
| TVD | 4 | 38.734 | 36.152 | 43.07 | 38.586 | 53.885 | 50.363 | 58.606 | 53.563 |
| | 16 | 54.616 | 54.361 | 55.715 | 54.04 | 70.042 | 69.269 | 71.184 | 69.881 |
| | 64 | 60.655 | 60.356 | 60.243 | 58.753 | 74.583 | 74.319 | 74.778 | 74.093 |
| MSE | 4 | 29.297 | 27.906 | 38.862 | 34.722 | 37.697 | 36.982 | 49.464 | 47.08 |
| | 16 | 44.487 | 45.211 | 50.327 | 49.023 | 53.266 | 53.791 | 60.958 | 60.297 |
| | 64 | 51.391 | 51.739 | 55.02 | 53.845 | 58.854 | 59.171 | 64.686 | 64.191 |

| Settings | Denoising steps | Quasar-T METEOR $\gamma=1.4$ | $\gamma=2.0$ | ROUGE-L $\gamma=1.4$ | $\gamma=2.0$ | QQP METEOR $\gamma=1.4$ | $\gamma=2.0$ | ROUGE-L $\gamma=1.4$ | $\gamma=2.0$ |
|---|---|---|---|---|---|---|---|---|---|
| Vanilla-fine-tuned (teacher) | 4 | 21.777 | 18.493 | 26.671 | 21.272 | 32.724 | 31.472 | 34.202 | 31.086 |
| | 16 | 26.731 | 25.34 | 31.699 | 28.33 | 38.341 | 38.251 | 38.679 | 37.202 |
| | 64 | 28.237 | 27.663 | 33.307 | 30.795 | 40.017 | 40.068 | 39.882 | 38.686 |
| mode-seeking KL | 4 | 22.523 | 19.525 | 27.836 | 22.476 | 33.71 | 32.989 | 35.124 | 32.653 |
| | 16 | 27.224 | 26.303 | 32.59 | 29.522 | 38.978 | 39.28 | 39.252 | 38.153 |
| | 64 | 28.684 | 28.636 | 34.014 | 32.047 | 40.523 | 40.921 | 40.349 | 39.489 |
| mode-covering KL | 4 | 22.133 | 19.253 | 27.271 | 22.29 | 33.085 | 32.19 | 34.614 | 31.853 |
| | 16 | 27.146 | 26.074 | 32.278 | 29.386 | 38.749 | 39.006 | 39.122 | 37.905 |
| | 64 | 28.49 | 28.628 | 33.672 | 31.899 | 40.354 | 40.764 | 40.26 | 39.349 |
| TVD | 4 | 22.461 | 19.292 | 27.746 | 22.352 | 33.594 | 32.602 | 35.079 | 32.236 |
| | 16 | 27.249 | 26.364 | 32.584 | 29.654 | 38.896 | 39.197 | 39.214 | 38.062 |
| | 64 | 28.619 | 28.606 | 33.973 | 31.958 | 40.442 | 40.874 | 40.314 | 39.414 |
| MSE | 4 | 20.39 | 15.045 | 27.728 | 21.364 | 27.398 | 27.984 | 30.943 | 29.878 |
| | 16 | 25.005 | 20.196 | 31.837 | 26.088 | 31.116 | 31.972 | 33.892 | 33.507 |
| | 64 | 26.281 | 21.963 | 33.045 | 27.679 | 31.845 | 33.241 | 34.448 | 34.548 |

trajectory-distilled MDLM from round $r-1$. We show the plots on two different guidance scales, considering that we distill on a range of guidance scales.

**Our distilled models achieve up to $16\times$ speedup** in conditional generation while maintaining competitive quality. Across multiple benchmarks, we observe substantial efficiency gains with guidance- and trajectory-distilled MDLMs. On the Bible style transfer (PUB-BBE and PUB-ASV) tasks, our 7-round distilled MDLM with only 16 denoising steps matches the METEOR and ROUGE-L scores of the vanilla-fine-tuned teacher at 128 denoising steps (i.e., 256 NFEs). This translates to a $16\times$ overall speedup ($2\times$ from guidance distillation and $8\times$ from trajectory distillation). Similar improvements are observed for paraphrasing (QQP) and question generation (Quasar-T) tasks. Beyond this, our proposed method also enhances *few-step generation*, compared to the vanilla-fine-tuned MDLM, e.g., on the QQP dataset, our student MDLM with $\gamma = 2.0$ achieves a **+14.4** point gain in METEOR scores and **+11.7** point gain in ROUGE-L scores over the teacher MDLM for 2-step generation. Moreover, performance improves progressively across rounds: as the student learns to approximate the teacher over larger step sizes, the student's few-step-generation capabilities steadily strengthen. This trend is evident in our main results, where intermediate-round models that already offer strong efficiency–quality trade-offs are further improved in later rounds.

**Guidance-distilled MDLMs achieve performance that matches, or even surpasses, the performance of their teachers.** This comes as an added benefit along with the improved efficiency

**Figure 2:** Sample generated outputs from the public Bible versions→basic English (top) and QQP paraphrasing (bottom) tasks. NFE refers to the number of function evaluations performed to generate the output. We write "NFE $= 2 \times N$" to indicate the use of $N$ denoising steps with 2 NFEs per step for classifier-free guided generation. Generation errors are indicated in red.

---

**Public Bible Versions → Bible in Basic English (PUB-BBE)**

**Input:** So the priest gave him hallowed bread: for there was no bread there but the shewbread, that was taken from before the Lord, to put hot bread in the day when it was taken away.

**Generated Outputs:**

- **Vanilla-fine-tuned (NFE = 2 × 128):** So the priest gave him holy bread: for there was no bread there but the holy bread which had been taken away from before the Lord, to put warm bread on the day when it was taken away.

- **Vanilla-fine-tuned (NFE = 2 × 16):** So the priest gave him holy bread: for there was no bread there only only holy bread which had been taken away before before the Lord, to put warm bread on the day when it has taken taken taken away.

- **Vanilla-fine-tuned (NFE = 2 × 4):** So the priest gave him holy bread; him there because there no only no bread bread bread, the before which the before before before before before before before before the, on the the when the had taken taken taken taken taken taken taken.

- **Distilled (NFE = 16):** So the priest gave him the holy bread: for there was no bread there but the holy bread which was taken from before the Lord, to put soft bread on the day when it was taken away.

- **Distilled (NFE = 4):** So the priest gave him a holy bread: for there was no bread there but the holy bread bread was taken taken before before the Lord, to put salt bread on the day when it was taken away

---

**Sentence Paraphrasing (QQP)**

**Input:** What will be the general equation of the following curve?

**Generated Outputs:**

- **Vanilla-fine-tuned (NFE = 2 × 128):** What is the general equation of the following curve?

- **Vanilla-fine-tuned (NFE = 2 × 16):** What is the general equation of the following simple curve curve?

- **Vanilla-fine-tuned (NFE = 2 × 4):** What is the general equation of following curve curve curve curve curve curve curve??

- **Distilled (NFE = 16):** What is the general equation of the following curve?

- **Distilled (NFE = 4):** What is the general equation of the following curve?

---

obtained by the reduction of two forward passes to a single pass for the computation of the guided output distribution. To better understand this, we conduct **ablation studies** on guidance distillation using the following loss functions:

1. **Mode-seeking KL divergence:** $\sum_x D_{\mathrm{KL}}\big(p_{\mathrm{student}}(x) \,\|\, p_{\mathrm{teacher}}(x)\big)$,

2. **Mode-covering KL divergence:** $\sum_x D_{\mathrm{KL}}\big(p_{\mathrm{teacher}}(x) \,\|\, p_{\mathrm{student}}(x)\big)$,

3. **Total Variation Distance (TVD):** $(1/2) \sum_x \| p_{\mathrm{teacher}}(x) - p_{\mathrm{student}}(x) \|_1$, and

4. **Mean Squared Error (MSE):** $\sum_x (1/|x|) \| \log p_{\mathrm{teacher}}(x) - \log p_{\mathrm{student}}(x) \|_2^2$.

Table 1 shows the performance with the aforementioned guidance-distillation loss functions. Our results show that MDLMs distilled with divergence-based objectives (KL divergence and TVD) perform comparably, often outperforming the vanilla-fine-tuned teacher MDLM, with the mode-seeking variant of KL divergence yielding the best performance. We attribute this to its *zero-forcing* property: the loss imposes stronger penalties when the student places probability mass in locations (over vocabulary) where the teacher has lower likelihood. This behavior suppresses the sampling of low-probability tokens during generation. By contrast, the MSE loss underperforms substantially, even falling short of the vanilla-fine-tuned teacher model. This highlights the critical role of divergence-based training objectives in effective guidance distillation.

**Qualitative analysis:** Figure 2 compares generations from the vanilla-fine-tuned teacher model and our distilled student model on PUB-BBE style transfer and paraphrasing tasks. The qualitative observations are consistent with the quantitative results. For the teacher model, we observe a sharp degradation in sample quality as the NFEs decrease. This degradation arises because reducing NFEs forces the MDLM to predict/unmask larger sets of tokens independently at each step, weakening the dependencies among them and leading to incoherent sequences. In contrast, the distilled model exhibits substantially better few-step generation, retaining high quality even with as low as 16 NFEs; while an initial speedup is achieved by imitating guided outputs in a single step, major improvement stems from transferring trajectory knowledge from the teacher, enabling the student to learn stronger token-level dependencies and form more coherent associations across the sequence. More generated samples appear in Appendix E.

## 6 RELATED WORK

**Diffusion language modeling:** Inspired by diffusion generative modeling for images (Ho et al., 2020; Song et al., 2021a;b), diffusion language modeling has recently surfaced as a non-autoregressive alternative to the traditional next-word prediction language modeling paradigm. Prior works have explored Gaussian diffusion process (similar to image diffusion modeling) in word embedding spaces for language modeling (Li et al., 2022) and also explored its scaling laws (Gulrajani & Hashimoto, 2023). Works of Gong et al. (2023); Lin et al. (2022); Yuan et al. (2024) have explored diffusion modeling for seq-to-seq NLP tasks with encoder-decoder based architectures. Our work is based on diffusion modeling in discrete spaces (Hoogeboom et al., 2021; Austin et al., 2021; Zheng et al., 2024). Specifically, we work with masked or absorbing-state discrete diffusion modeling (Sahoo et al., 2024; Shi et al., 2024) where the diffusion noising process iteratively masks out the tokens in the sentence. Recent works of Nie et al. (2025b), Zhu et al. (2025), and Ye et al. (2025) have shown the scalability of masked diffusion for language modeling, showcasing competitive performance to autoregressive LLMs.

**Diffusion distillation:** Distillation of diffusion models trained to generate images has been widely explored in the literature. Early works of Luhman & Luhman (2021) distilled by generating a synthetic dataset from a pre-trained diffusion model using deterministic sampling. Later works of Salimans & Ho (2022) and Berthelot et al. (2023) bypassed the creation of a synthetic dataset by progressive multi-round distillation of diffusion models to reduce the number of generation steps. Based on these works, Meng et al. (2023) proposed distillation of guided image diffusion models; this is also the most relevant work to our research. Consistency distillation (Song et al., 2023) presents distillation techniques for one-step image generation. For MDLMs, the progressive distillation strategy called self-distillation through time (SDTT) has been proposed by Deschenaux & Gulcehre (2025). This work was followed by Hayakawa et al. (2025), which introduces dimensional correlations to further distill the models trained using SDTT. For uniform diffusion language models (UDLMs), the work of Sahoo et al. (2025) connects UDLMs to Gaussian diffusion and proposes discrete consistency distillation inspired by consistency models in image generation.

## 7 SUMMARY, CONCLUSION AND FUTURE WORK

In this work, we present distillation methods for conditional masked diffusion language models for efficient inference on seq-to-seq NLP tasks. Our methods tackle two main causes of sampling inefficiencies, i.e., (i) classifier-free guidance and (ii) multi-step denoising generation, reducing the total number of function evaluations required for generation without any significant degradation in generation quality. Empirically, we show a speedup of up to $16\times$ on various seq-to-seq NLP tasks by reducing the number of function evaluations while virtually retaining the performance. By improving efficiency and few-step sample quality, our proposed method significantly enhances the capability of MDLMs for task-specific practical use cases. Our proposed dual-distillation framework shows great potential along two promising directions. First, our framework can be extended to *multimodal MDLMs* and to larger model architectures, broadening its applicability. Second, more exploration can be conducted to leverage structured correlations across token dimensions (Hayakawa et al., 2025) within conditional settings where dimensional dependencies are crucial.

## 8 REPRODUCIBILITY STATEMENT

The training experiments are performed on top of the open-sourced models provided by Nie et al. (2025a). The general training and inference settings are mentioned in Section 4.2. More details on hyperparameters for the specific tasks and distillation stages are mentioned in Appendix D. Additional details regarding the implementation have been mentioned in Appendix C. We plan to open-source our code and trained model weights upon de-anonymization.

## 9 ETHICS STATEMENT

Generative language models are trained on large-scale datasets that may contain societal biases, which can inadvertently be reflected in the model outputs. Moreover, such models can be misused for unethical purposes, including generating misleading content or facilitating plagiarism. The focus of this work is on improving the efficiency of masked diffusion language models, and as such, it does not introduce additional ethical risks beyond those already associated with generative language models. Nevertheless, we acknowledge the broader ethical considerations surrounding the deployment of generative models and emphasize the importance of responsible use.

## 10 STATEMENT OF LARGE LANGUAGE MODELS USAGE

Large Language Models (LLMs) were used solely as an assistive tool for improving the clarity and readability of the manuscript. No part of the research ideation, methodology design, experimental execution, or analysis was conducted using LLMs. All refinements suggested by the LLM were carefully reviewed and verified by the authors prior to inclusion. The authors take full responsibility for the final content of this paper.

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

# APPENDIX

## A  CONSTRUCTING TARGETS FOR TRAJECTORY DISTILLATION

We construct a distillation target distribution for trajectory distillation by solving a two-step reverse diffusion process from a randomly sampled $t \sim \mathcal{U}(0,1)$ and a fixed step size $\Delta$. To do so, we defined a function `solve` that takes a noisy input $\boldsymbol{x}_t$, step size, and the teacher denoiser. Algorithm 3 describes the `solve` function in detail. For solving a single step, we follow the reverse diffusion equations from Equation 2 and obtain a sample $\boldsymbol{x}_{t-\Delta}$. On the positions that got unmasked in the step, we collect the log-probabilities predicted by the teacher denoiser. We perform the reverse solving step and log-probabilities step again, obtaining a sample $\boldsymbol{x}_{t-2\cdot\Delta}$. Finally, we perform a forward pass through the teacher denoiser (i.e., a prediction at $t = 0$ from a sample at timestep $t - 2 \cdot \Delta$) and collect log-probabilities on all the remaining masked positions in $\boldsymbol{x}_{t-2\cdot\Delta}$. In this way, we obtain the training target log-probabilities on all masked positions of $\boldsymbol{x}_t$.

---

**Algorithm 3:** `solve`

---

**Given:** Noisy sequence $\boldsymbol{x}_t$ of length $L$, step size $\Delta$, denoiser $\widehat{\mathbf{x}}_\theta$, vocabulary size $V$, minimum possible noise level $\epsilon(> 0)$.

$\text{out} \leftarrow \mathbf{0} \in \mathbb{R}^{L \times V}$

**for** $k = 1, 2$ **do**

    $p_\theta^\gamma(\mathbf{x}_0|\boldsymbol{x}_t, \boldsymbol{y}) \leftarrow \exp\left(\widehat{\mathbf{x}}_\theta(\boldsymbol{x}_t, \boldsymbol{y}, \gamma)\right);$

    $\boldsymbol{x}_0 \sim p_\theta^\gamma(\mathbf{x}_0|\boldsymbol{x}_t, \boldsymbol{y})$ ;           ▷ Sample $\boldsymbol{x}_0$ from the denoiser predictions

    $s \leftarrow \max(t - k \cdot \Delta, \epsilon)$ ;     ▷ Next step in reverse based on the step size $\Delta$

    **for** *all positions $i$ in the sequence* **do**

        **if** $x_t^i \neq \langle m \rangle$ **then**

            $x_s^{(i)} \leftarrow x_t^{(i)}$ ;                 ▷ Umasked tokens remain unchanged

        **else**

$$x_s^{(i)} \leftarrow \begin{cases} \langle m \rangle & \text{with probability } \frac{s}{t}, \\ x_0^{(i)} & \text{with probability } 1 - \frac{s}{t} \end{cases} ; \quad \triangleright \text{ Follows from Equation 2}$$

        **if** $x_s^{(i)} \neq x_t^{(i)}$ **then**

            $\text{out}^{(i)} \leftarrow \log p_\theta^\gamma(\mathbf{x}_0^{(i)}|\mathbf{x}_t, \mathbf{y})$ ;     ▷ Collect log-probs on unmasked positions

    $\boldsymbol{x}_t \leftarrow \boldsymbol{x}_s$

**for** *$j$ where $x_t^{(j)} = \langle m \rangle$* **do**

    $\text{out}^{(j)} \leftarrow \widehat{\mathbf{x}}_\theta^{(j)}(\boldsymbol{x}_t, \boldsymbol{y}, \gamma)$ ;     ▷ Collect log-probs on the remaining masked positions after simulating two reverse steps

**return** `out`

---

## B  MASKED DIFFUSION TRAINING AND INFERENCE

Algorithm 4 outlines the training procedure for MDLMs. Given a training instance $x_0$, a random timestep $t$ is sampled, after which the forward masked diffusion process is applied. The model is then trained to reconstruct the masked positions using cross-entropy loss. Finally, the gradients of this loss are computed and used to update the model parameters via standard optimization.

Algorithm 5 presents the sampling procedure for MDLMs. In this work, we adopt the greedy decoding strategy introduced by Chang et al. (2022). At each timestep, let $k$ denote the number of tokens to be unmasked. The algorithm selects the $k$ tokens with the highest predicted probabilities under $p_\theta(\mathbf{x}_0|\boldsymbol{x}_t, \boldsymbol{y})$. Furthermore, the prediction distribution $p_\theta(\mathbf{x}_0|\boldsymbol{x}_t, \boldsymbol{y})$ may be enhanced using classifier-free guidance, as described in Section 3.1.

---

**Algorithm 4:** MDLM Training Algorithm

---

**Given:** MDLM Denoiser $\widehat{\mathbf{x}}_\theta(\cdot, \boldsymbol{y})$, input/source sentence $\boldsymbol{y}$, mask token $\langle m \rangle$.

**repeat**

    $(\boldsymbol{x}_0, \boldsymbol{y}) \sim$ data;

    $t \sim \mathcal{U}(0,1)$;

    $\boldsymbol{x}_t \sim q_{t|0}(\mathbf{x}_t | \boldsymbol{x}_0)$ ;                ▷ forward diffusion process

    $\log p_\theta(\mathbf{x}_0 | \mathbf{x}_t, \mathbf{y}) \leftarrow \widehat{\mathbf{x}}_\theta(\boldsymbol{x}_t, \boldsymbol{y})$ ;     ▷ forward pass through the network

    $\mathcal{L}_{\mathrm{MDM}} = - \sum\limits_{x_t^{(i)} = \langle m \rangle} \log p_\theta(\mathbf{x}_0^i | \boldsymbol{x}_t, \boldsymbol{y})$ ;     ▷ Compute loss

    Gradient descent with $\nabla_\theta \mathcal{L}_{\mathrm{MDM}}$

**until** *convergence*;

---

**Algorithm 5:** MDLM Sampling Algorithm

---

**Given:** Trained MDLM denoiser $\widehat{\mathbf{x}}_\theta(\cdot, \boldsymbol{y})$, input/source sentence $\boldsymbol{y}$, Total denoising steps $= T$, generation sequence length $L$, mask token $\langle m \rangle$.

**Define:** $c_i$ as the confidence of a token $\boldsymbol{x}_0^i$ as assigned by the distribution $p_\theta(\mathbf{x}_0^i | \boldsymbol{x}_t, \boldsymbol{y})$.

$\boldsymbol{x}_t \leftarrow \langle m \rangle_{|L|}$ ;          ▷ Initialize as a sequence of $\langle m \rangle$ tokens.

**for** $t = 1, \frac{T-1}{T}, \frac{T-2}{T}, ..., \frac{1}{T}$ **do**

    $s = t - \frac{1}{T}$;

    $k = \lfloor L(1-s) \rfloor$ ;       ▷ number of unmasked tokens at timestep $s$

    $p_\theta(\mathbf{x}_0 | \boldsymbol{x}_t, \boldsymbol{y}) \leftarrow \widehat{\mathbf{x}}_\theta(\boldsymbol{x}_t, \boldsymbol{y})$ ;     ▷ forward pass through the network

    $\boldsymbol{x}_0 \sim p_\theta(\mathbf{x}_0 | \boldsymbol{x}_t, \boldsymbol{y})$;

    **for** $i = [1, 2, ..., L]$ **do**

        **if** $x_t^i \neq \langle m \rangle$ **then**

            $x_0^i = x_t^i$ and $c_i = 1$ ;     ▷ Retained already unmasked tokens

        **else**

            $x_0^i \sim p_\theta(\mathbf{x}_0^i | \boldsymbol{x}_t, \boldsymbol{y})$ and $c_i = p_\theta(\mathbf{x}_0^i = x_0^i | \boldsymbol{x}_t, \boldsymbol{y})$

        **if** $c_i \in$ *top-k* $\left( \{c_l\}_{i=1}^{L-1} \right)$ **then**

            $x_s^i = x_0^i$ ;    ▷ Unmask tokens with top-k highest confidence of

            sampling

    $\boldsymbol{x}_t \leftarrow \boldsymbol{x}_s$;

return $\mathbf{x}_0$

---

## C   Architecture and Implementation Details

Since we use the pre-trained models from Nie et al. (2025a), we adopt their Transformer encoder as the backbone architecture for MDLMs. In contrast to standard diffusion model architectures, no additional timestep embeddings are incorporated. Our model is configured with 12 layers, a hidden size of 768, 12 attention heads, and an intermediate dimension of 3072. For the distilled MDLMs, we have guidance scale as an extra input to the network (as mentioned in Section 3.1). The scalar guidance scale is first converted to a sinusoidal embedding similar to positional embeddings used in Transformer-based language models. This sinusoidal embedding goes through a two-layer MLP with SiLU activation function, and outputs a 768-dimensional embedding vector. We treat this vector corresponding to the guidance scale as a token embedding and prepend it to the embedding sequence of the input tokens for conditioning.

**Implementation:** The input to the denoiser is given by concatenating the conditional sequence $\boldsymbol{y}$ and the noisy input $\boldsymbol{x}_t$. Consider the conditional sequence $\boldsymbol{y} = (y^{(0)}, y^{(1)}, y^{(2)}, \ldots, y^{(L_1-1)})$ and the ground truth target sequence $\boldsymbol{x}_0 = (x_0^{(0)}, x_0^{(1)}, x_0^{(2)}, \ldots, x_0^{(L_2-1)})$. The input to the denoiser after the forward process will look something like $\boldsymbol{x}_{\mathrm{inp}} = (y^{(0)}, y^{(1)}, y^{(2)}, \ldots, y^{(L_1-1)}, \langle \mathrm{sep} \rangle, x_0^{(0)}, \langle m \rangle, x_0^{(2)}, \ldots, \langle m \rangle)$, where $\langle \mathrm{sep} \rangle$ is a separator token or string. The ground truth sequence will be partially (or fully) masked depending on the noise

| Task | Setup | LR | warmup steps | batch size | Max sequence length | Training steps | steps per round | Initial step size |
|---|---|---|---|---|---|---|---|---|
| **PUB-BBE and PUB-ASV** | Vanilla-fine-tuning | 3e-4 | 500 | 256 | 192 | 22.5K | - | - |
| | Guidance distillation | 7e-5 | 1000 | 192 | | 25K | - | - |
| | Trajectory distillation | 8e-5 | 500 | 160 | | 70K | 10K | 1/128 |
| **QQP** | Vanilla Fine-tuning | 1e-5 | 1000 | 512 | 64 | 20K | - | - |
| | Guidance distillation | 5e-5 | 500 | 512 | | 36K | - | - |
| | Trajectory distillation | 1e-5 | 100 | 512 | | 30K | 5K | 1/64 |
| **Quasar-T** | Vanilla fine-tuning | 5e-6 | 1000 | 256 | 64 | 16K | - | - |
| | Guidance distillation | 5e-6 | 500 | 256 | | 33.5K | - | - |
| | Trajectory distillation | 1e-5 | 100 | 256 | | 25K | 5K | 1/32 |

**Table 2:** Hyperparameter settings for the fine-tuning and distillation training on the tasks performed.

level randomly selected during the training step (see Algorithm 4). After $x_{\text{inp}}$ is passed through the denoiser, we gather the logit predictions at the masked positions and ignore the unmasked positions. The denoiser learns to predict masked tokens given the unmasked ones. During inference, we use the trained network to predict all masked positions, keeping predictions on the unmasked tokens intact. In distillation, the objective changes to mimicking the distribution of tokens on the masked positions output by the trained teacher denoiser. As mentioned in Section 4.2, we maintain a set of parameters with an exponential moving average (EMA) with some decay $d_{\text{ema}}$. The way to do this is to first initialize the EMA parameters $\theta_{\text{ema}}$ with the parameters of the denoiser $\theta$ before training. The optimization/parameter update happens as follows:

$$\theta \leftarrow \theta - \kappa \cdot \nabla \mathcal{L};$$
$$\theta_{\text{ema}} \leftarrow (1 - d_{\text{ema}}) \cdot \theta + d_{\text{ema}} \cdot \theta_{\text{ema}}. \tag{5}$$

Here, $\kappa$ is the learning rate. The exponential moving average of parameters stabilizes the parameter update and helps generalize better. To enable classifier-free guidance, Ho & Salimans (2021) proposes to train the denoiser on conditional and unconditional generation in tandem by randomly dropping out the condition with a small probability during training. However, with MDLMs, Nie et al. (2025a) shows that the unconditional outputs can be obtained by simply masking out the conditional sequence. This doesn't need any training-time dropping of the conditional sequence, as the unconditional setting in MDLMs (i.e., the conditional sequence being masked out) gets implicitly learned.

To decode which positions to re-mask and which to retain, we greedily choose the tokens predicted with the highest probability (Chang et al., 2022). The ordering of unmasking can be influenced by designing different inference-time heuristics (see Algorithm 5); however, we do not explore them in this work.

## D HYPERPARAMETER DETAILS

Table 2 provides the hyperparameter values used during training under different datasets and settings. For PUB-BBE and PUB-ASV, we use the entire training corpus of (Carlson et al., 2017), which consists of paired data between different versions of the Bible. So, we train a single model on the entire dataset and evaluate on the task-specific test sets. The common training and inference settings are mentioned in Section 4.2.

## E  ADDITIONAL GENERATED OUTPUTS

---

**MDLM outputs from PUB-BBE**

**Input:** and your fields, and your vineyards, and your olive-yards – the best – he doth take, and hath given to his servants.

**Generated Outputs:**

- **Vanilla-fine-tuned (NFE $= 2 \times 128$):** And let him take your fields and your vine-gardens and your olive-gardens, the best, and let him give to his servants.

- **Vanilla-fine-tuned (NFE $= 2 \times 16$):** And his fields take your best vines and vine-ens and your olivelive-treesens, and give to his servants as servants servants

- **Vanilla-fine-tuned (NFE $= 2 \times 4$):** And your fields take and your vine-gardens and your o–liveliveens heives and and his to give his servants servants

- **Distilled (NFE=16):** And your fields and your vine-gardens and your olive-gardens he will take and given it to his servants.

- **Distilled (NFE=4):** And your fields and your vine-ardens and your olive-gardens, he will take and and give to his servants

---

**MDLM outputs from PUB-ASV**

**Input:** And Yahweh said to him, "But I will be with you, and you will defeat Midian as if they are one man.

**Generated Outputs:**

- **Vanilla-fine-tuned (NFE $= 2 \times 128$):** And Jehovah said unto him, But I will be with thee, and thou shalt smite Midian, as though they were one man.

- **Vanilla-fine-tuned (NFE $= 2 \times 16$):** And Jehovah said unto him, But I will be with thee, and thou shalt sm Midian as though though were were one man man

- **Vanilla-fine-tuned (NFE $= 2 \times 4$):** And Jehovah said unto him, But I will be with thee, and thou shalt smite Midian as as though they as as.

- **Distilled (NFE=16):** And Jehovah said unto him, But I will be with thee, and thou shalt overcome Midian as though they be one man.

- **Distilled (NFE=4):** And Jehovah said unto him, But I will be with thee, and thou shalt strike Midian as though they be one man.

---

**MDLM outputs from QQP**

**Input:** How do I convert this 35 mm film camera to a digital camera from scratch?

**Generated Outputs:**

- **Vanilla-fine-tuned (NFE $= 2 \times 128$):** How do I convert a 35 mm film to a digital camera?

- **Vanilla-fine-tuned (NFE $= 2 \times 16$):** How do I convert a 35 mm film camera to digital camera camera?

- **Vanilla-fine-tuned (NFE $= 2 \times 4$):** How do I convert film cameramm film camera camera camera camera camera camera camera

- **Distilled (NFE=16):** How do I convert a 35 mm film camera to a digital camera?

- **Distilled (NFE=4):** How can I convert a 35 mm film camera to a digital camera?

---

**MDLM outputs from Quasar-T**

**Input:** Rather than using pure egg tempera , which is dry pigment ground in water and mixed with an egg yolk for the underpainting , Gnidziejko uses an egg-oil emulsion which consists of the pigment and water.

**Generated Outputs:**

- **vanilla-fine-tuned (NFE = 2 × 128):** Which painting medium consists of egg yolks and water?
- **vanilla-fine-tuned (NFE = 2 × 16):** Which painting medium is in water and egg yolks water?
- **vanilla-fine-tuned (NFE = 2 × 4):** Which painting medium involves water water egg egg egg egg egg egg yol
- **Distilled (NFE=16):** Which painting medium uses egg yolks and water?
- **Distilled (NFE=4):** Which painting medium involves water egg and and

