# OpenReview forum: "Dual Distillation of Trajectory and Guidance Knowledge for Faster Inference in Conditional Masked Diffusion Language Models"
_ICLR.cc/2026/Conference — ICLR 2026 Conference Withdrawn Submission_

### Official Review · Reviewer_Ag1V · 2025-10-15

**Soundness:** 2
**Presentation:** 3
**Contribution:** 2
**Rating:** 2
**Confidence:** 3

**Summary:**

This paper proposes a two-stage distillation method to address the high inference cost of conditional MDLMs. Evaluated on multiple tasks, the approach achieves up to $16\times$ speedup ($2\times$ from guidance distillation and up to $8\times$ from trajectory distillation) while maintaining or improving output quality. Qualitative examples indicate that the distilled model performs markedly better in low-step regimes, producing more coherent outputs with fewer denoising steps. The authors further suggest that the framework can be extended to multimodal MDLMs and scaled to larger architectures, potentially broadening its applicability.

**Strengths:**

1. The paper is well-written and easy to follow, with clear explanations of the two-stage distillation methodology and comprehensive algorithmic descriptions.

2. Under the authors' experimental settings, the two-stage approach achieves substantial speedup across multiple standard benchmarks while maintaining generation quality. The qualitative analysis with generated examples further supports this claim.

**Weaknesses:**

1. Limited Model Scale and Task Scope. Experiments are conducted only on a 113M parameter model. Scaling to larger MDLMs such as LLaDA [1] or Dream [2] remains unverified, despite the authors' claim that the framework can be extended to larger architectures. Furthermore, evaluation is restricted to relatively simple sequence-to-sequence tasks. Generalization to other conditional generation tasks (e.g., mathematical and code generation) is unclear. If the authors could provide supplementary results demonstrating the framework's acceleration potential on larger models and on challenging benchmarks (e.g., GSM8K [3], MATH [4], HumanEval [5]), the scalability claims would be significantly more convincing.

2. The claims regarding CFG acceleration are questionable. As shown in Table 1 of SMDM [6], the model can still achieve good results without using CFG. In the authors' claimed $16\times$ speedup, $2\times$ comes from stage-one guidance distillation. However, for the vanilla fine-tuned teacher model, using CFG may not be necessary. If the authors could provide evaluation results without CFG, it would further confirm the necessity of using CFG and better demonstrate the utility of guidance distillation.



[1] Nie et al., Large Language Diffusion Models.

[2] Ye et al., Dream 7B: Diffusion Large Language Models.

[3] Cobbe et al., Training Verifiers to Solve Math Word Problems.

[4] Hendrycks et al., Measuring Mathematical Problem Solving With the MATH Dataset.

[5] Chen et al., Evaluating Large Language Models Trained on Code.

[6] Nie et al., Scaling up Masked Diffusion Models on Text.

**Questions:**

1. The paper states that for guidance distillation, $\gamma_{\min} = 1$ and $\gamma_{\max} = 3$ are chosen, and results are reported for $\gamma = 1.4$ and $\gamma = 2.0$. What is the rationale behind these specific choices? Additionally, in Table 2, each task uses different initial step sizes and total numbers of rounds. How were these values determined, and why is a unified set of hyperparameters not used across all tasks?

2. Could the authors provide a comparison with the concurrent work D2F [1]? Such a comparison would help contextualize the contributions of the proposed approach.

[1] Wang et al., Diffusion LLMs Can Do Faster-Than-AR Inference via Discrete Diffusion Forcing.

---

> ### Author Response · Authors · 2025-11-23
>
> We thank the reviewer for the thoughtful feedback and constructive comments. We hope to address the raised concerns below:
>
> > Limited Model Scale and Task Scope.
>
> Our experiments utilize a 113M (non-embedding) parameter model, as this size enables a controlled evaluation of the dual-distillation framework. While we agree that demonstrating results on larger architectures and more challenging conditional tasks would further strengthen the work, our method operates at the level of the diffusion trajectory and guidance signals, and does not rely on architectural assumptions specific to small models. This is why we state that the framework is extensible, but we intentionally refrain from claiming empirical scalability without additional evidence. We will add a limitations section to clarify this and highlight scaling to larger MDLMs and more challenging benchmarks as a key direction for future work.
>
> > The claims regarding CFG acceleration are questionable.
>
> Classifier-free guidance (CFG) has become a very standard technique in diffusion models [1], used in almost every conditional setting to control the trade-off between diversity and fidelity. As mentioned in the paper, using CFG often improves performance if the guidance scale is chosen well, making its usage a crucial part of the sampling process. The only drawback is that computing CFG outputs requires two forward passes per denoising step. Our proposed method eliminates this requirement. We urge the reviewer to examine Figure 3a from [2] to gain a better understanding of this for the Bible style transfer task. In the figure, a guidance scale of 1.0 indicates no CFG, and it is clear that “no CFG” clearly underperforms and does not scale well with increasing denoising steps.
>
> > The paper states that for guidance distillation, $\gamma_{\min} = 1$ and $\gamma_{\max} = 3$ are chosen, and results are reported for $\gamma = 1.4$ and $\gamma = 2.0$. What is the rationale behind these specific choices?
>
> We apologise for not being clear enough on the choices made. Initially, we observed that the performance gains from increasing the guidance scale $\gamma$ plateaued at a value near $\gamma=3$. For the min value, i.e., $ \gamma=1$, this represents the case of no CFG. In the evaluation phase, we simply showed two different guidance scale values to demonstrate performance improvements in the selected range. We will update the paper clarifying these choices.
>
> > Additionally, in Table 2, each task uses different initial step sizes and total numbers of rounds. How were these values determined, and why is a unified set of hyperparameters not used across all tasks?
>
> The reason some tasks use a different set of initial step sizes and total rounds is that we observed that for QQP and QG, we could gain essentially the same observations with fewer distillation rounds. This saves training time for these tasks.
>
> > Could the authors provide a comparison with the concurrent work D2F [1]?
>
> We appreciate the suggestion. The work of D2F[3] aims to achieve a similar goal of inference-time acceleration. However, the work of D2F builds upon block diffusion [4], which makes MDLMs semi-autoregressive, allowing the use of KV caching to speed up inference. While it is possible to extend our methodology to block diffusion and D2F, our goal was to tackle inference-time inefficiencies arising from the CFG and denoising steps, whereas block diffusion and D2F focus on inference-time inefficiencies brought about by redundant computation. Although the goal is the same (i.e., to reduce inference-time inefficiencies), the work is orthogonal to ours and does not form an appropriate baseline.
>
> **References:**
>
> [1] Sadat et al., No Training, No Problem: Rethinking Classifier-Free Guidance for Diffusion Models.
>
> [2] Padole et al., Improving Text Style Transfer using Masked Diffusion Language Models with Inference-time Scaling.
>
> [3] Wang et al., Diffusion LLMs Can Do Faster-Than-AR Inference via Discrete Diffusion Forcing.
>
> [4] Arriola et al., Block Diffusion: Interpolating Between Autoregressive and Diffusion Language Models

---

> > ### Comment · Reviewer_Ag1V · 2025-11-24
> >
> > I thank the authors for their rebuttal. The authors have addressed my concerns regarding the effectiveness for CFG and the specific hyperparameter settings.
> >
> > I acknowledge the authors' clarification that the proposed method operates at the level of diffusion trajectories and is technically architecture-agnostic. However, given that the experiments are limited to a 113M parameter model, the demonstrated improvements may be bounded, and the evidence provided is not fully persuasive regarding the method's broader impact. While I accept that scaling to larger architectures is listed as future work, I maintain a reservation regarding the **scalability and potential** of the proposed method when applied to larger MDLMs (e.g., LLaDA, Dream) and more standard, challenging benchmarks.
> >
> > In light of the authors' response, I have raised my score to 4 accordingly.

---

### Official Review · Reviewer_y9eZ · 2025-10-19

**Soundness:** 2
**Presentation:** 3
**Contribution:** 3
**Rating:** 4
**Confidence:** 4

**Summary:**

This paper proposes a two-stage knowledge distillation framework for accelerating inference in conditional MDLMs on sequence-to-sequence NLP tasks. The first stage CFG into a single forward pass, eliminating the computational overhead of computing both conditional and unconditional distributions. The second stage progressively distills multi-step denoising into fewer steps through self-distillation. The approach achieves up to 16× speedup (2× from guidance distillation, 8× from trajectory distillation) while maintaining generation quality across three tasks.

**Strengths:**

1. The paper addresses genuine computational bottlenecks in MDLMs—both the dual forward passes required by CFG and the multi-step generation process. These are real inefficiencies affecting practical deployment. And efficiency of the advantage of MDLM over AR.
2. The two-stage distillation approach is straightforward and well-explained. The architectural modification (adding a guidance scale embedding) for encoding guidance preferences is simple and practical.

**Weaknesses:**

1. While aims to improve the sampling efficiency of MDLM, this work only presents experiments with three specific conditional text generation, unlike many existing work that address the unconditional generation and more challenging tasks.
2. Limited technical novelty: The paper essentially applies existing techniques (guidance distillation from Meng et al. 2023, progressive distillation from Salimans & Ho 2022 and the discrete progressive distillation SDTT from Deschenaux & Gulcehre 2025) to conditional MDLMs. While the adaptation is competent, the core methodological contribution is incremental. The authors acknowledge this is an extension of prior image diffusion work but position it as the "first" for guided conditional MDLMs on seq-to-seq tasks, which is a narrow novelty claim.
3. There are many missing baselines in the direction of KV cache and faster sampler for MDLM

**Questions:**

$q_0$ is missing in the expectation in eq(3)

---

> ### Author Response · Authors · 2025-11-23
>
> We thank the reviewer for the thoughtful feedback and constructive comments. We hope to address the raised concerns below:
>
> > While aims to improve the sampling efficiency of MDLM, this work only presents experiments with three specific conditional text generation
>
> We agree that broader evaluations would further strengthen the work. One of the aims of our paper was to eliminate the requirement of two forward passes per denoising step to compute classifier-free guided outputs. To demonstrate this, it is necessary to evaluate conditional generation and not rely solely on unconditional generation as demonstrated in [2]. In addition to that, our current experiments use a small-scale 114M (non-embedding) parameter model (167M total), and therefore focus on seq-to-seq tasks where the teacher itself is reliable enough to meaningfully assess distillation gains. For more complex domains such as math or code, the teacher’s limited capability would confound the evaluation of acceleration and quality improvements. Our goal in this submission is to establish the feasibility and benefits of dual distillation in a controlled setting; scaling the approach to larger teachers and harder reasoning tasks is a natural next step. We will clarify this design choice and include an explicit limitations section in the revised paper.
>
> > Limited technical novelty
>
> While our framework builds upon diffusion distillation methods in image generation [1] and unconditional text generation [2] (as we acknowledge in the paper), our contribution lies in the application to a different modality and the choice of tasks. Although the technical novelty appears limited, distilling conditional masked diffusion LMs introduces fundamentally different training dynamics; the model must distill both guidance and trajectory under discrete masked-token corruption rather than Gaussian noise. These differences require further study, such as the proposed dual distillation tailored to masked-token transitions, which are absent in prior work.
>
> > There are many missing baselines in the direction of KV cache and faster sampler for MDLM
>
> Training-free acceleration methods, such as KV-caching and faster sampler for MDLM, target orthogonal aspects of efficiency and can, in fact, be combined with our approach. KV-caching remains applicable after distillation and would provide additional speedups independent of the quality improvements obtained through dual distillation. Similarly, a faster sampler for MDLM (for instance, the work on entropy-bounded unmasking by [3]) is an alternative decoding strategy to the confidence-based scheme we use, rather than a competing distillation method. Since these techniques serve different purposes, they do not form a direct point of comparison.
>
> > $q_0$ is missing in the expectation in eq(3)
>
> We thank the reviewer for bringing this to our attention. The term $q_0$ should indeed be included in the expectation for completeness. We omitted it for brevity, but agree that adding it improves clarity and correctness. We will update the equation in the revised manuscript.
>
> **References:**
>
> [1] Meng et al. On Distillation of Guided Diffusion Models.
>
> [2] Deschenaux et al. Beyond Autoregression: Fast LLMs via Self-Distillation Through Time.
>
> [3] Ben-Hamu et al. Accelerated Sampling from Masked Diffusion Models via Entropy Bounded Unmasking.

---

> > ### Comment · Reviewer_y9eZ · 2025-11-25
> >
> > I am still not convinced by the rebuttal from the authors, especially in term of the scope of the experiments and the significancy of the technique novelty.
> >
> > Could the authors explain further in terms of the training dynamics?
> > > distilling conditional masked diffusion LMs introduces fundamentally different training dynamics.
> >
> > Also I do not understand why the authors want to mention the Gaussian noise, if we are talking about MDLM and conditional MDLM:
> > >  the model must distill both guidance and trajectory under discrete masked-token corruption rather than Gaussian noise.
> >
> > Given the above considerations, I tend to maintain the score at 4.

---

### Official Review · Reviewer_G1S3 · 2025-10-23

**Soundness:** 2
**Presentation:** 3
**Contribution:** 1
**Rating:** 4
**Confidence:** 4

**Summary:**

This paper addresses the high computational cost of conditional masked diffusion language models (MDLMs) with distillation techniques. MDLMs have the advantage of parallel (non-autoregressive) generation but require running many denoising (unmasking) steps during inference, and the classifier-free guidance (CFG) procedure even doubles the number of network evaluations. The paper proposes a two-stage distillation algorithm for conditional MDLMs. In the first stage, a student MDLM, which takes the guidance strength $\gamma$ as an extra input, is trained to emulate the output of a CFG-guided teacher. In the second stage, the guidance-distilled student model from Stage 1 is then used as the initial teacher for a series of progressive distillation steps, where each step trains a new student to distill two teacher unmasking steps into a single step. Empirical results demonstrate that the final student model achieves accelerated inference speed while maintaining comparable performance on NLP seq-to-seq generation tasks.

**Strengths:**

1. This paper identifies two sources of inefficiency in conditional MDLMs (iterative unmasking and CFG) and adapts existing diffusion distillation techniques to address these issues.
2. Empirical results and ablation studies on NLP seq-to-seq generation tasks demonstrate that the distilled models achieve 16x speedup at inference time without significant degradation of generation quality.
3. The paper is generally well-written and easy to understand.

**Weaknesses:**

1. The novelty of the proposed method is limited, as the proposed dual distillation framework is a direct adaptation of existing distillation method in the literature. This contribution does not meet the acceptance bar in my opinion.
2. Although the proposed method achieves a good speedup at inference time, it involves a computationally very expensive distillation procedure (especially in stage two where several rounds of progressive distillation are performed), which limits the scalability of the proposed method for larger models and larger datasets.
3. Limitations are not discussed in the paper.

**Questions:**

1. How is the diversity of the generated sequences from the final student model compared to those from the initial teacher model?
2. How is the inference speed of the final student model compared to standard auto-regressive LLMs of similar size?
3. Given the significant training costs introduced by the proposed method, could the authors quantify the total training cost and elaborate on this trade-off?
4. In the second stage, the proposed method adapts progressive distillation, which requires several rounds of distillation to produce a multi-step student model. However, recent diffusion distillation approaches (e.g., [1,2,3, 4]) can distill diffusion models into one-step student models in one round. Could those approaches be adapted to the conditional MDLM distillation setting to reduce training costs?
5. Please add a paragraph to discuss the limitations of the proposed method in the paper.
6. Is Equation (2) based on the assumption that $\alpha_t=1-t$? This assumption cannot be found in Sec 2.1.

[1] S Xie, et al. "EM distillation for one-step diffusion models." NeurIPS 2024.

[2] W Luo, et al. "Diff-Instruct: A universal approach for transferring knowledge from pre-trained diffusion models." NeurIPS 2023.

[3] M Zhang, et al. "Towards Training One-Step Diffusion Models Without Distillation". arXiv.

[4] H Zheng, et al. "Ultra-fast language generation via discrete diffusion divergence instruct." arXiv.

---

> ### Author Response · Authors · 2025-11-23
>
> We thank the reviewer for the thoughtful feedback and constructive comments. We hope to address the raised concerns below:
>
> > The novelty of the proposed method is limited
>
> While our framework builds upon diffusion distillation methods in image generation [1] and unconditional text generation [2], our contribution lies in the application to a different modality and the choice of tasks. Distilling conditional masked diffusion LMs introduces fundamentally different training dynamics; the model must distill both guidance and trajectory under discrete masked-token corruption rather than Gaussian noise. These differences require further study, such as the proposed dual distillation tailored to masked-token transitions, which are absent in prior work.
>
> > Although the proposed method achieves a good speedup at inference time ....
>
> Discussed below addressing the 3rd question raised by the reviewer.
>
> > Limitations are not discussed in the paper.
>
> We apologise for this oversight. We will update the paper with a dedicated limitations section.
>
> > How is the diversity of the generated sequences ....
>
> We compute the diversity of the generated outputs by sampling 8 sentences for each input. The metrics used are self-BLEU and div-4, widely used in the literature. Preliminary results on the PUB-BBE and PUB-ASV tasks are presented below.
>
> **PUB-BBE:**
>
> | Model              | NFE | Self-BLEU | Div-4 |
> |--------------------|-----|-----------|-------|
> | Vanilla fine-tuned | 128 | 0.881     | 0.360 |
> | Final Distilled    | 128 | 0.917     | 0.299 |
> | Final Distilled    | 16   | 0.591     | 0.673 |
>
> **PUB-ASV:**
>
> | Model              | NFE | Self-BLEU | Div-4 |
> |--------------------|-----|-----------|-------|
> | Vanilla fine-tuned | 128 | 0.916     | 0.317 |
> | Final Distilled    | 128 | 0.938     | 0.262 |
> | Final Distilled    | 16  | 0.863     | 0.413 |
>
> > How is the inference speed of the final student model compared to standard auto-regressive LLMs of similar size?
>
> To compare the inference speed between our MDLM (114M non-embedding params, 167M total), we train and evaluate using SmolLM2 [3] (135M params in total). For fair comparison, we use the same batch size (i.e., 128) during generation and compute tokens per second generated by these models. For AR sampling, we use top-p sampling with $ p = 0.95$. Preliminary results on PUB-BBE are given below.
>
> | Model            | Parameters | NFE | TPS     | ROUGE-L |
> |------------------|------------|-----|---------|---------|
> | AR (SmolLM2)     | 135M       | -   | 7763.12 | 42.964  |
> | MDLM (Distilled) | 167M       | 16  | 7137.71 | 61.107  |
> | MDLM (Distilled) | 167M       | 128 |  908.68 | 61.547  |
>
> With distillation, we reach similar inference speeds as an AR model of similar size with better performance. We will further extend these experiments to other tasks and add them to the final version of the paper.
>
> > Given the significant training costs introduced ....
>
> As mentioned in the paper, all experiments have been conducted on NVIDIA RTX A6000 GPUs. In Table 2 of the Appendix, we outline the exact training steps required for all settings. To give an idea of the time taken, the distillation experiments require approximately 16 hours of training time for trajectory distillation on the Bible style transfer task, which is the maximum time among all the tasks performed. This is about 4 times higher than the vanilla fine-tuning cost. As a trade-off for this one-time increase in training cost, we observe a 16 times reduction in the number of function evaluations required during inference time.
>
> > In the second stage, the proposed method adapts progressive distillation, which requires several rounds of distillation to produce a multi-step student model. However ....
>
> We agree with the reviewer's suggestion. In fact, a parallel work is currently under review in ICLR 2026, which can be found here: https://openreview.net/forum?id=inXScPz1gT. The work adapts consistency models for image generation to discrete diffusion settings and provides a promising direction (similar to what the reviewer has mentioned). Our framework can be potentially improvised in this direction in the future, following this line of research.
>
> > Is Equation (2) based on the assumption that $\alpha_t=1-t$? This assumption cannot be found in Sec 2.1.
>
> We apologise for this oversight. The reviewer is correct that the assumption is $\alpha_t = 1-t$. We will explicitly add this assumption and update the loss function accordingly. For readers seeking a more general formulation, we will direct them to the work of [4] in the paper.
>
> **References:**
>
> [1] Meng et al., On Distillation of Guided Diffusion Models.
>
> [2] Deschenaux et al., Beyond Autoregression: Fast LLMs via Self-Distillation Through Time.
>
> [3] Allal et al., SmolLM2: When Smol Goes Big -- Data-Centric Training of a Small Language Model.
>
> [4] Sahoo et al., Simple and Effective Masked Diffusion Language Models.

---

### Official Review · Reviewer_ai6w · 2025-11-02

**Soundness:** 2
**Presentation:** 3
**Contribution:** 2
**Rating:** 4
**Confidence:** 4

**Summary:**

This paper introduces a two-stage distillation framework for conditional MDLMs to improve inference efficiency in sequence-to-sequence NLP tasks. The method distills (i) classifier-free guidance and (ii) unmasking trajectory knowledge from a teacher MDLM into a student model. The distilled student can generate text with a single forward pass per step and fewer unmasking steps, achieving up to 16× speedup while maintaining comparable generation quality

**Strengths:**

1. The paper is well written and clearly motivated; both the overall idea and the two-stage distillation algorithm are clearly presented and easy to follow.

2. Experiments on sequence-to-sequence tasks convincingly demonstrate the effectiveness of the proposed method in improving inference efficiency while maintaining generation quality.

**Weaknesses:**

1. The novelty of this work appears limited. The two proposed algorithms mainly follow the frameworks of previous studies [1, 2], and the paper does not clearly explain how its approach differs conceptually or technically from them.

2. The experimental scope is narrow. The paper focuses on relatively simple seq-to-seq tasks, where the reported inference acceleration is not particularly impressive. It would be more convincing to include evaluations on challenging tasks such as mathematical reasoning or code generation, which are central to current research on large language models and diffusion language models [3, 4].

[1] Meng et al. On Distillation of Guided Diffusion Models.

[2] Deschenaux et al. Beyond Autoregression: Fast LLMs via Self-Distillation Through Time.

[3] Ye at al. Dream 7B: Diffusion Large Language Models.

[4] Xie et al. Dream-coder 7b: An open diffusion language model for code.

**Questions:**

How does the proposed method compare with training-free acceleration approaches [5, 6] that are widely adopted in the community?

[5] Wu et al. Fast-dLLM: Training-free Acceleration of Diffusion LLM by Enabling KV Cache and Parallel Decoding.

[6] Ben-Hamu et al. Accelerated Sampling from Masked Diffusion Models via Entropy Bounded Unmasking.

---

> ### Author Response · Authors · 2025-11-23
>
> We thank the reviewer for the thoughtful feedback and constructive comments. We hope to address the raised concerns below:
>
> > The novelty of this work appears limited.
>
> While our framework builds upon diffusion distillation methods in image generation [1] and unconditional text generation [2] (as we acknowledge in the paper), our contribution lies in the application to a different modality and the choice of tasks. Distilling conditional masked diffusion LMs introduces fundamentally different training dynamics; the model must distill both guidance and trajectory under discrete masked-token corruption rather than Gaussian noise. These differences require further study, such as the proposed dual distillation tailored to masked-token transitions, which are absent in prior work. We will revise the manuscript to clarify these distinctions more explicitly.
>
> > The experimental scope is narrow.
>
> We agree that broader evaluations would further strengthen the work. Our current experiments use a small-scale 114M (non-embedding) parameter model (167M total), and therefore focus on seq-to-seq tasks where the teacher itself is reliable enough to meaningfully assess distillation gains. For more complex domains such as math or code, the teacher’s limited capability would confound the evaluation of acceleration and quality improvements. Our goal in this submission is to establish the feasibility and benefits of dual distillation in a controlled setting; scaling the approach to larger teachers and harder reasoning tasks is a natural next step. We will clarify this design choice and include an explicit limitations section in the revised paper.
>
> > How does the proposed method compare with training-free acceleration approaches that are widely adopted in the community?
>
> We appreciate the suggestion. Training-free acceleration methods, such as KV-caching and entropy-bounded unmasking, target orthogonal aspects of efficiency and can, in fact, be combined with our approach. KV-caching remains applicable after distillation and would provide additional speedups independent of the quality improvements obtained through dual distillation. Similarly, entropy-bounded unmasking is an alternative decoding strategy to the confidence-based greedy sampling (proposed in [3]) that we use, rather than a competing method. Since these techniques serve different purposes, they do not form a direct point of comparison.
>
> **References:**
>
> [1] Meng et al. On Distillation of Guided Diffusion Models.
>
> [2] Deschenaux et al. Beyond Autoregression: Fast LLMs via Self-Distillation Through Time.
>
> [3] Chang et al. MaskGIT: Masked Generative Image Transformer

---

### Note · Authors · 2026-01-19

**Comment:**

We sincerely thank the reviewers for their time and effort in evaluating our manuscript, as well as for their thoughtful and constructive feedback. We appreciate the detailed comments and valuable suggestions offered across the reviews. After carefully considering the feedback and noting several common concerns and directions for improvement raised by multiple reviewers, we have decided to withdraw the paper from ICLR 2026. We believe that addressing these points will significantly strengthen both the technical quality and the overall presentation of the work. We therefore plan to thoroughly revise the manuscript, incorporating the reviewers’ suggestions, and hope to submit an improved version in the future. We are grateful to the reviewers and the program committee for their consideration of our work.

**Withdrawal Confirmation:**

I have read and agree with the venue's withdrawal policy on behalf of myself and my co-authors.